# Dynamics of Maize Grain Weight and Quality during Field Dehydration and Delayed Harvesting

Rulang Zhao [1,2,3,†], Yonghong Wang [3,†], Xiaofang Yu [1], Wanmao Liu [2,4], Daling Ma [1], Hongyan Li [2], Bo Ming [2], Wenjie Zhang [3], Qiming Cai [3], Julin Gao [1,*] and Shaokun Li [2,*]

1    Agricultural College, Inner Mongolia Agricultural University, Hohhot 010018, China; langzi9620@126.com (R.Z.); yuxiaofang75@163.com (X.Y.); madaling@sina.com (D.M.)
2    Institute of Crop Sciences, Chinese Academy of Agricultural Sciences, Beijing 100081, China; liuwm@nxu.edu.cn (W.L.); lhy200_1@163.com (H.L.); obgnim@163.com (B.M.)
3    Institute of Crop Sciences, Ningxia Academy of Agriculture Sciences, Yinchuan 750002, China; wyhnx2002-3@163.com (Y.W.); zwj870131@163.com (W.Z.); cqmahz189@163.com (Q.C.)
4    College of Agronomy, Ningxia University, Yinchuan 750021, China
*    Correspondence: nmgaojulin@163.com (J.G.); lishaokun@caas.cn (S.L.); Tel./Fax: +86-0471-4302403 (J.G.); +86-10-82108891 (S.L.)
†    These authors contributed equally to this work.

**Abstract:** Dehydration of maize grains in the field can reduce costs associated with drying after harvest. A delayed harvest approach after physiological maturity, in which plants stand in the field to allow the stems to dry, has been widely adopted in maize production. However, it remains unclear how harvesting at different time points during the dehydration stage may affect grain yield and quality. In the present study, experiments were conducted in the Ningxia Irrigation Area of northwest China from 2019 to 2022, we continuously observed and used a linear-plateau model to analyze the changes in grain weight and quality traits (such as bulk density and levels of starch, protein, oil, fiber, and free fatty acids) during the field dehydration period of maize hybrids with differing maturity times. Harvesting at a grain moisture content of >31.0% was shown to affect grain weight, whereas harvesting at a moisture content of <25.9% did not affect grain weight or yield. The stable period for grain weight occurred during the physiological maturity stage for an early-maturing hybrid and 5–12 days before physiological maturity for the mid–late-maturing hybrids. When the field grain moisture content was <37%, harvesting did not affect the grain bulk density. Grain bulk density tended to stabilize one to two weeks earlier than grain weight and two to three weeks before the physiological maturity period. The protein, oil, fiber, and free fatty acid contents in maize kernels at 30 days after silking were not affected by the harvesting period, and the starch contents were unaffected in maize kernels harvested at any time later than 50 days after silking. Overall, maize grain should be harvested during field dehydration and delayed harvesting after physiological maturity with relatively low moisture content to get a better yield with superior quality.; delayed harvesting is therefore an important technical approach to improve the efficient production of high-quality maize.

**Keywords:** maize; dehydration period; moisture content; kernel weight; bulk density; grain quality; harvest time

## 1. Introduction

Maize (*Zea mays* L.) is the most abundantly produced food crop throughout the world. A wide variety of cultivars are planted, most of which produce high yields, and there are extensive, well-established maize industrial chains across the globe [1–4]. In addition, maize has important roles as a feed material in aquaculture and as a raw material in industry. It is a highly significant target for ensuring food security and feed safety in China [5–7]. The past few decades have seen significant changes in Chinese maize harvesting methods, beginning with traditional manual harvesting, transitioning to mechanical ear harvesting,

then progressing to mechanical grain harvesting [8–10]. Mechanical harvesting from plants with low moisture content not only reduces drying costs and improves harvesting efficiency, but also improves maize quality. It is therefore an important method in modern maize production [8,11].

The dehydration process for maize grains in the field includes dehydration both before and after physiological maturity (PM) [12,13]. Delayed harvest after PM is a universal management strategy that is used to reduce grain drying costs worldwide [14–16]. Delaying harvest, drying straw in the field, dehydrating maize grains, and directly harvesting and storing grains only after the moisture content (MC) has been reduced to a safe storage level greatly reduces drying costs [11,15]. Delayed (i.e., low-moisture) harvesting of maize takes advantage of regional light and heat resources, and is an important future direction in maize production [8,17,18]. However, significant controversy remains regarding the impacts of delayed harvest on maize yield. Some studies have shown that yield losses are caused by delayed harvesting after PM [19,20], whereas other studies have found no significant changes in grain weight during the straw-drying period in the field after PM [12,21–23]. However, reductions in grain MC resulting from delayed harvesting have been demonstrated consistently [14,24].

Grain MC is an important factor in crop harvesting, storage, and preservation management and a key characteristic that determines grain pricing [25,26]. Grain traders are willing to pay higher prices for maize with MC < 13.0% (which can be immediately stored safely) than for grains with unknown or higher MC [27]. In the northern Midwest region of the USA, the cost of grain drying is a major expenditure in maize production, second only to fertilizer or seed purchases [14,28]. The standard MC range for grain sales and safe storage is 13–15.5% [21]. High MC in stored grains can lead to aflatoxin contamination; the fungi that produce aflatoxins can breed in the grains when MC exceeds 13% [29,30]. The Harvest Quality Report by the US Food Commission showed that the average grain MC was 16.41% during the maize harvest period from 2011 to 2019 [18]. Maize MC during the PM period typically ranges from 15–42%, with significant differences between plants with different genetic backgrounds and growth environments [8,31].

Grain quality is another key economic trait that is crucial in maize production and directly determines the market price and application value of maize [32]. Maize grains with high bulk density are generally considered to be high-quality [33]. The bulk density and nutritional composition of maize kernels form the basis of the metabolic energy supply and nutritional value of feed products [34]. A study of changes in the nutritional quality of mature maize found no significant changes in protein, oil, or starch content in maize kernels after PM or during 1–2 weeks of delayed harvesting, indicating that maize can be harvested at any time after PM [14]. Previous studies have revealed structural changes in starches and proteins in the endosperm during the drying process of maize kernels; such changes primarily depend on environmental temperatures and the grain MC [35,36]. Previous studies of maize kernel quality have focused on the effects of various environmental factors (such as temperature and light), cultivation measures, maize varieties, fertilizer regimens, and water management measures [37–40]. There has been comparatively little research into changes in maize grain quality as a result of harvesting at different points during the dehydration period.

The Ningxia Irrigation Area belongs to China's high-yield maize area. This region has abundant light and heat resources, and there is just one maize growing season per year. Thus, the abundance of light and heat resources both before and after the PM period are underutilized by the plants. A study has not yet been conducted to establish the critical time points associated with grain weight and to assess key quality traits during the maize stem dehydration period in the field. Such data would provide important theoretical support for the formulation of technical measures to regulate grain quality and accelerate dehydration before PM. Furthermore, these data would allow the promotion and application of delayed, low-MC grain harvesting technologies in other regions of the world with similar environments.

Our hypothesis was that field dehydration and delayed harvesting affected the weight and quality of the grains significantly. Therefore, the objectives of this study were as follows: (1) to elucidate the dynamic changes in grain weight and key quality indicators during the field dehydration period; (2) to identify the time point after which harvesting would not affect maize grain weight or key grain quality indicators; and (3) to comprehensively evaluate the impacts of low-moisture harvesting on maize grain quality. These results will provide a theoretical basis for the implementation of delayed low-moisture maize harvesting in similar areas in northwest China. Overall, this study has important guiding significance for promoting the development of a high-quality maize industry in this region and throughout the world.

## 2. Materials and Methods

### 2.1. Experimental Site

All experiments were conducted at the Wanghong Experimental Base of the Crop Research Institute, Ningxia Academy of Agricultural and Forestry Sciences, Yongning County, Ningxia Irrigation District, China (116°41′ E, 39°91′ N) from 2019 to 2022. Ningxia Irrigation District includes the Yang-Huang Irrigation District and the Yellow River Diversion Irrigation District. This region has a typical continental semi-humid and semi-arid climate and is located within the latitude zone containing high-yield maize fields in China [41]. The region has an altitude of 1100–1400 m. The total solar radiation was 1692.3 W/m$^2$; the sunshine duration was 3000 h/year; the average annual temperature was 8–9 °C; the annual accumulated temperature (AT) $\geq$ 10 °C was 3100–3300 °C; and the average sunshine percentage was >60% [42]. The annual rainfall was 200–300 mm, with increases in precipitation corresponding to increases in altitude from the north to the south of the region. The average daily temperature from late September to late October in each year from 2019–2022 was 15.3 °C, 13.2 °C, 10.5 °C, and 8.1 °C, respectively; the 10-d rainfall averages during the same periods were 11.2 mm, 7.4 mm, 3.9 mm, and 4.1 mm, respectively [11] (Figure 1). The average physical and chemical properties of the soil at the test site were assessed at a depth of 0–40 cm (Table 1).

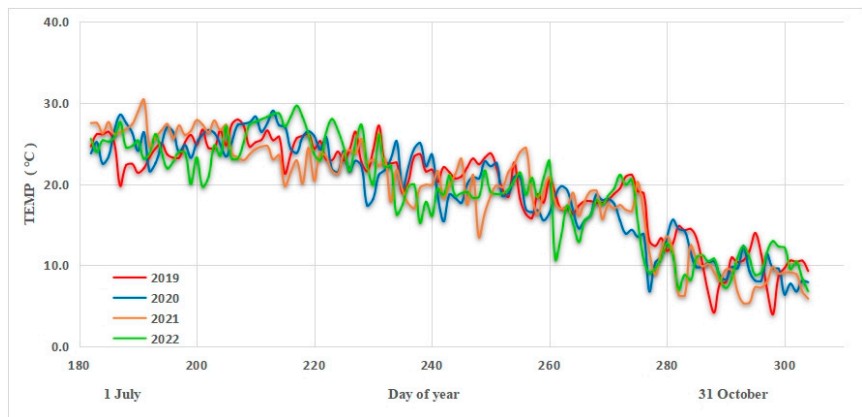

**Figure 1.** Average daily temperatures during the period from silking to field dehydration in 2019 through 2022.

**Table 1.** The average Physical and chemical properties of test site soil at a depth of 0–40 cm.

| pH | Organic Matter (g kg$^{-1}$) | Total N (g kg$^{-1}$) | Available N (mg kg$^{-1}$) | Available P (mg kg$^{-1}$) | Available K (mg kg$^{-1}$) |
| --- | --- | --- | --- | --- | --- |
| 7.87 | 15.11 | 0.84 | 54.84 | 21.81 | 99.57 |

### 2.2. Experimental Design and Field Management

Four hybrids that are widely planted in Northwest China were selected, namely the early-maturing hybrid KWS9384 and three mid–late-maturing hybrids (DK159, XY335, and YY439). The key growth periods and AT ≥ 0 °C from the silking to PM period are detailed in Table 2. The planting density was 90,000 plants ha$^{-1}$, with each hybrid growing in an area >667 m$^2$. Other field management conditions were consistent with local practices. The average temperature and other meteorological data required for calculating AT were obtained from the small meteorological station installed in the experimental field (Watch Dog 2900 Weather Station).

### 2.3. Sampling and Measurements

Measurements were taken during the entire process of grain filling and dehydration in the early-maturing and the mid–late-maturing maize hybrids, ranging from 30 d after silking until 30 d after PM. Measurements were taken every 10 d, and it is ~5 d before PM. If precipitation occurred on a scheduled sampling day, sampling was postponed until the following day. Take 5 consecutive spikes between rows of each hybrid each time, 5 consecutive spikes are taken from the rows of each hybrid, along with the bracts. Samples were returned to the laboratory for manual threshing. From the middle of each ear, 100 kernels were removed and the fresh weight (*FW*) was measured. The kernels were then dried in an air-drying oven at 85 °C for 48 h before the dry weight (*DW*) was measured. The grain *MC* was calculated from the kernel *FW* and *DW* as follows [43]:

$$MC(\%) = \frac{FW - DW}{FW} \times 100\% \tag{1}$$

To accurately predict the optimal harvest period for each cultivar based on the grain *MC*, we analyzed the relationship between *AT* and *MC* after the silking stage [44–46]. A regression model was established to estimate the relationship between silking AT and MC [47]. Based on the distributions of *MC* and *AT*, a logistic power nonlinear growth model was selected to establish the regression models:

$$MC_P = \frac{90}{1 + \left(\frac{AT}{b}\right)^c} \tag{2}$$

where $MC_P$ is the predicted grain *MC* (%), *b* and *c* are regression parameters, and *AT* (≥ 0 °C) is the cumulative temperature (°C per d) starting from silk spinning. When the average air temperature was below 0 °C, this equation was not applicable.

Grain weight was defined as the DW of the middle grain of the ear. Grain bulk density (the mass of a given volume of grain, including air space) was measured for all four varieties in 2021 and 2022. The grain MC of each hybrid ranged from 10.4% to 12.2%. After the maize grains were dried in the shade, bulk density was measured in kg/hl using a GHCS-172 bulk density instrument (Shanghai Precision Instrumentation Co., Shanghai, China). For each cultivar, three samples of 1000 g were measured. Maize grain nutritional composition profiles were also established. Specifically, percentages of moisture, proteins, oils, starches, fiber, and free fatty acids were measured in dried samples with a DA7250 multifunctional full-spectrum near-infrared spectrum analyzer (Perten Instruments, Inc., Hägersten, Sweden). The nutritional component yield was calculated with the following equation [48]:

$$\text{Nutritional component yield} = \text{grain yield} \times \text{nutritional composition concentration (\%)} \tag{3}$$

The nutritional component yield and grain yield were expressed in mg ha$^{-1}$.

**Table 2.** Basic data for each hybrid tested in this study.

| Year | Hybrid Characters | | | | | | | | | |
|------|--------|-------------|-------------|----------------|--------------|----------|-------------------|------------------|-------------|------------------------------|
| | Hybrid | Hybrid Type | Sowing Date | Emergence Date | Silking Date | PM[a] Date | Growth Period (d) | SK[b]-PM (d) | SK-PM (°C) | Standing Straw after PM (d[c]) |
| 2019 | KWS9384 | EMV | 17 April | 1 May | 3 July | 31 August | 122 | 59 | 1491 | 64 |
| | XY335 | MLMV | 17 April | 1 May | 11 July | 22 September | 144 | 72 | 1731 | 39 |
| 2020 | KWS9384 | EMV | 21 April | 2 May | 2 July | 2 September | 123 | 62 | 1538 | 58 |
| | DK159 | MLMV | 21 April | 2 May | 10 July | 23 September | 144 | 75 | 1729 | 37 |
| | XY335 | MLMV | 21 April | 2 May | 10 July | 23 September | 144 | 75 | 1729 | 37 |
| | YY439 | MLMV | 21 April | 2 May | 10 July | 25 September | 146 | 77 | 1764 | 37 |
| 2021 | KWS9384 | EMV | 19 April | 3 May | 30 June | 29 August | 118 | 60 | 1444 | 56 |
| | DK159 | MLMV | 19 April | 3 May | 8 July | 21 September | 141 | 75 | 1676 | 35 |
| | XY335 | MLMV | 19 April | 3 May | 8 July | 21 September | 141 | 75 | 1676 | 35 |
| | YY439 | MLMV | 19 April | 3 May | 8 July | 24 September | 144 | 78 | 1729 | 32 |
| 2022 | KWS9384 | EMV | 22 April | 2 May | 1 July | 1 September | 122 | 61 | 1496 | 34 |
| | DK159 | MLMV | 22 April | 2 May | 10 July | 21 September | 142 | 73 | 1666 | 15 |
| | XY335 | MLMV | 22 April | 2 May | 10 July | 21 September | 142 | 73 | 1666 | 15 |
| | YY439 | MLMV | 22 April | 2 May | 10 July | 23 September | 144 | 75 | 1694 | 13 |

EMV, early-maturing hybrid; MLMV, mid–late-maturing hybrid; PM[a], physiological maturity; SK[b], silking date; d[c], days of standing in field after physiological maturity until harvest.

### 2.4. Statistical Analysis

Excel 2013 was used to calculate and plot the data. Origin 2022 was used to fit the data for the variation of maize grain moisture content and statistically analyze the data. Grain kernel weight, bulk density, nutritional composition and nutritional component yield were regressed against the day of the year using a linear-plateau model with the Origin 2022 (Piecewise Fit v1.40). The coefficient of determination ($R^2$) was significant at $p < 0.01$.

## 3. Results

### 3.1. Dynamic Changes in Grain Moisture during the Field Dehydration Period

Overall, grain MC decreased as the harvest period was delayed. There was a rapid decline period before PM followed by a slower decline after PM (Figure 2). There were differences in grain MC between hybrids and years during PM; the average grain MC in KWS9384, DK159, XY335, and YY439 was 29.9%, 27.2%, 24.7%, and 27.2%, respectively, during PM. The parameters used to fit the dehydration model for each cultivar are shown in Table 3. Among all of the tested hybrids, the coefficients of determination ($R^2$) between AT after silking and the grain MC ranged from 0.8747 to 0.9485, indicating that the models fit well. The predictive models of grain MC were used to calculate the time point at which MC would reach 25% in each cultivar. The early-maturing hybrid KWS9384 was predicted to reach 25% MC in early September, whereas the mid–late-maturing hybrids XY335, DK159, and YY439 were predicted to reach 25% MC at the end of September. Similarly, KWS9384 was predicted to reach 20% MC in mid-September, whereas XY335, DK159, and YY439 were predicted to reach 20% MC in early, mid-, and late October, respectively. The early-maturing hybrid KWS9384 dehydrated quickly after PM, and its MC was predicted to reach 16% by the end of September. XY335 was predicted to reach 16% MC in early November.

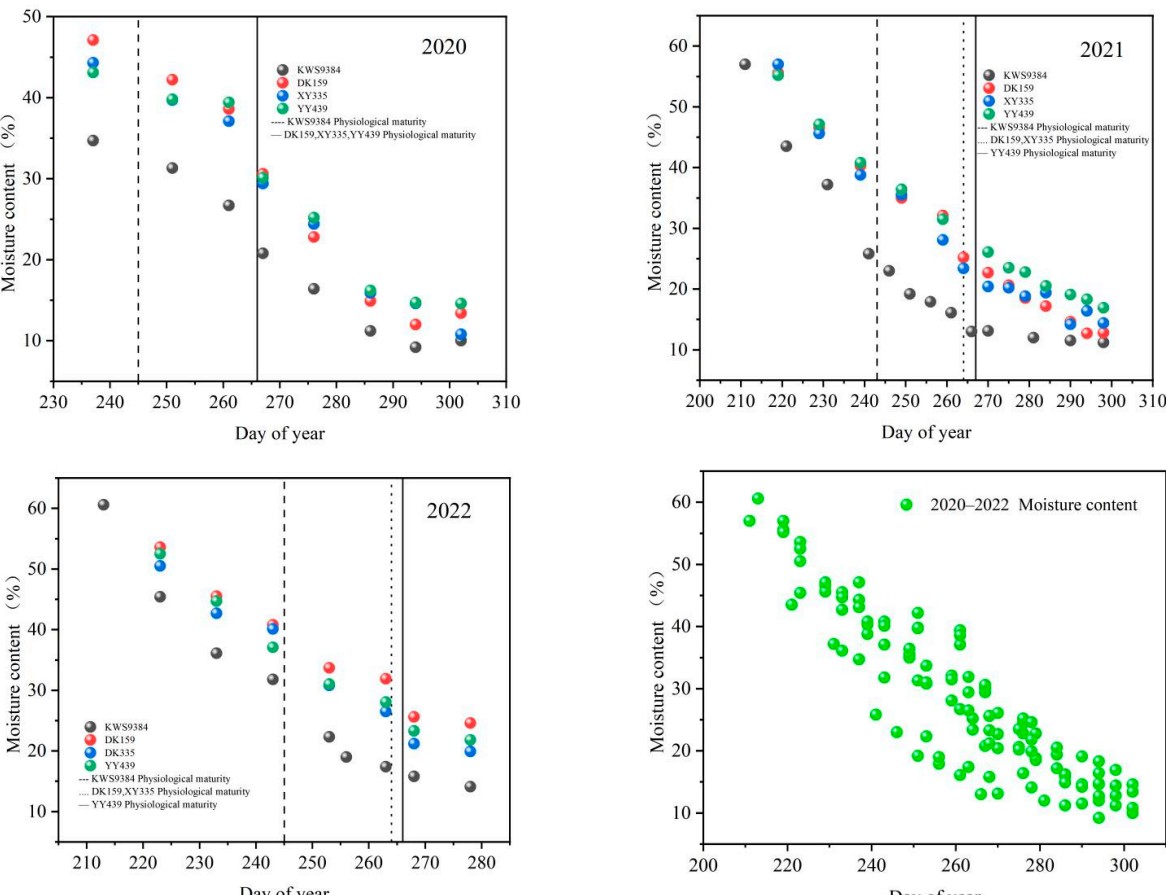

**Figure 2.** Changes in grain moisture content over the harvest period in four maize hybrids.

**Table 3.** Prediction of the harvest period for grains with different standard moisture contents and parameters used to fit grain dehydration models.

| Hybrid | MC% | | | | MC Standard | | | | | |
|---|---|---|---|---|---|---|---|---|---|---|
| | Model Coefficient | | | | 25% | | 20% | | 16% | |
| | b | c | R2 | MC of PM % | SK-25% AT °C | Date | SK-20% AT °C | Date | SK-16% AT °C | Date |
| KWS9384 | 1025.3 | 2.35 | 0.9485 ** | 29.9 | 1539.9 | 3 September | 1747.5 | 13 September | 1967.6 | 26 September |
| DK159 | 1139.1 | 2.32 | 0.8747 ** | 27.2 | 1720.9 | 24 September | 1956.5 | 10 October | 2206.8 | — — |
| XY335 | 1093.6 | 2.23 | 0.8950 ** | 24.7 | 1677.1 | 21 September | 1915.7 | 6 October | 2170.2 | 7 November |
| YY439 | 1106.3 | 2.03 | 0.8962 ** | 27.2 | 1771.2 | 26 September | 2050.4 | 20 October | 2352.1 | — — |

MC, moisture content. ** Coefficient of determination ($R^2$) was significant at $p < 0.01$. — — grain moisture content of this hybrid could not be predicted.

### 3.2. Changes in Kernel Weight during the Field Dehydration Period

The maize hybrids differed in the number of days required to reach PM. However, the pattern of kernel weight variation was the same for all tested hybrids, with the kernel weight stabilizing prior to PM. The 1000-kernel weight did not significantly change after PM for any of the hybrids. This pattern was consistent in 2021 and 2022. Overall, kernel weight first increased during the advanced harvest period, then stabilized. For each cultivar, this relationship was fitted to a linear-plateau model (Figure 3). The 1000-kernel weights of KWS9384, XY335, DK159, and YY439 stabilized at 0, 5, 9, and 12 days before PM, respectively. Based on the grain dehydration prediction model (Table 3), the grain MC values ranged from 25.9–31.0%, with an average of 28.7%, at the kernel weight stabilization point. In KWS9384, XY335, DK159, and YY439, the 1000-grain weight decreased by 4.44 g d$^{-1}$, 5.03 g d$^{-1}$, 5.57 g d$^{-1}$, and 5.6 g d$^{-1}$, respectively, before the kernel weight stabilized. In summary, when the grain MC was <31.0%, the harvest time did not affect the kernel weight.

### 3.3. Changes in Key Grain Qualities during the Field Dehydration Period

Bulk density is an important indicator for measuring grain quality and storage properties. Here, variations in bulk density were generally consistent with changes in kernel weight, and the patterns of variation in bulk density were consistent across hybrids. Furthermore, bulk density stabilized prior to PM. As with the 1000-kernel weight, grain bulk density showed a trend of first increasing, then stabilizing in all four hybrids. Stabilization tended to occur between 17 and 24 days before PM. The bulk density of the early-maturing hybrid KWS9384 decreased by 1.08 kg hl$^{-1}$ d$^{-1}$ in mid-August (17 days before PM), after which the harvest bulk density remained unchanged. For the mid–late-maturing hybrids DK159 and XY335, bulk density stopped increasing at the end of August (24 days before PM); earlier harvesting was associated with decreases of 0.65 kg hl$^{-1}$ d$^{-1}$ and 0.47 kg hl$^{-1}$ d$^{-1}$, respectively. In early September (22 days before PM), the early-harvest bulk density of the mid–late-maturing hybrid YY439 decreased by 0.43 kg hl$^{-1}$ d$^{-1}$. After this period, harvesting did not affect bulk density. Notably, the period of bulk density stability occurred earlier than grain weight stability (Figure 4). Based on the grain dehydration model, the predicted grain MC ranged from 37.1% to 41.3% when bulk density stabilized. Thus, harvesting maize at MC values of 25%, 20%, and 16% did not affect grain bulk density.

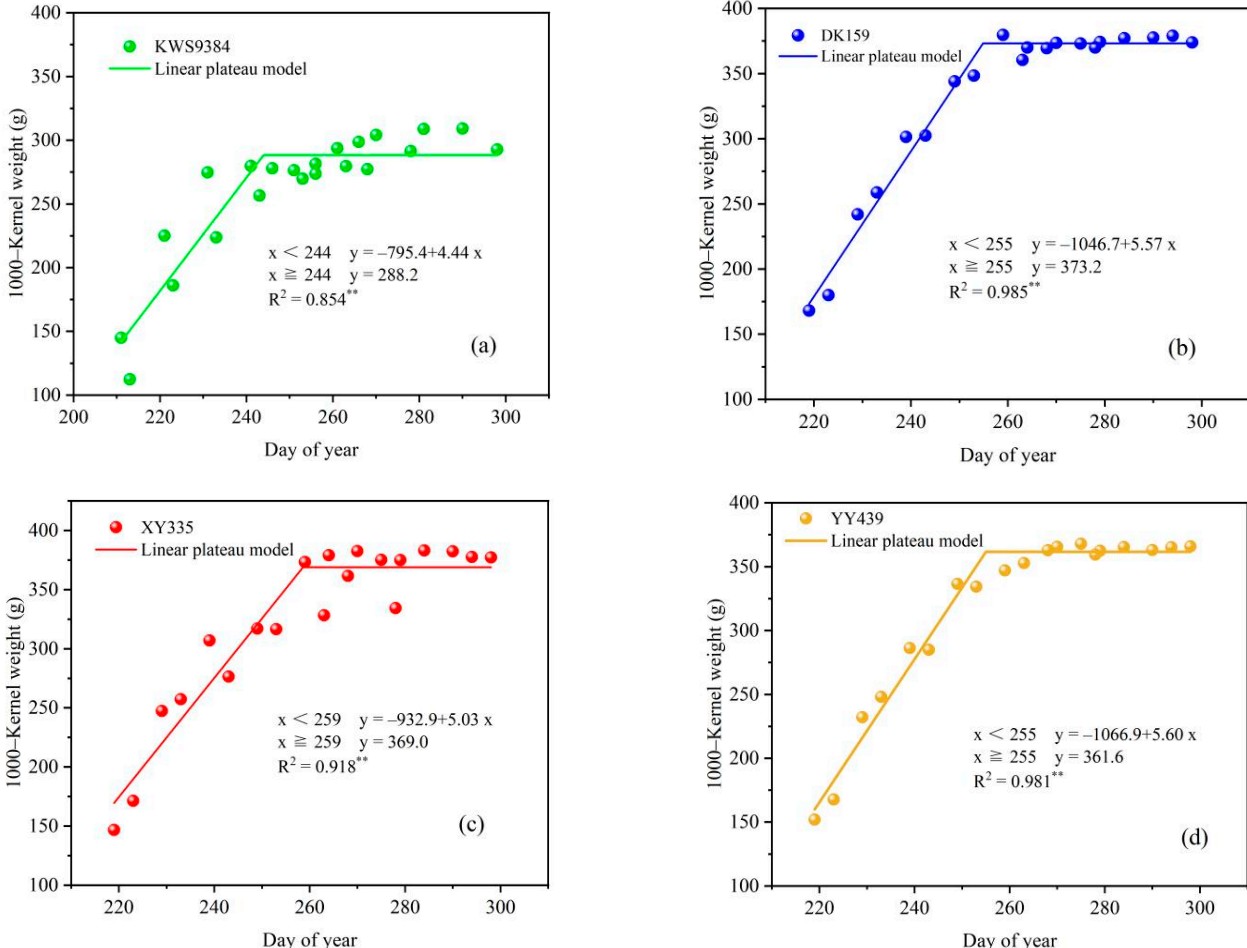

**Figure 3.** Maize 1000-kernel weight values during the harvest period for four hybrids. Linear-plateau models are shown for (**a**) KWS9384, (**b**) DK159, (**c**) XY335, and (**d**) YY439. ** Coefficient of determination (R2) was significant at *p* < 0.01.

There were no significant changes in the nutritional qualities of any hybrid from 30 days after silking until 30 days after PM. This indicated that maize could be harvested at any time beyond 30 days after silking and maintain optimal levels of protein, oil, fiber, and free fatty acids. The starch content showed a trend of first increasing, then stabilizing as the harvest period was delayed. The linear–plateau models for all four hybrids were statistically significant ($p$ < 0.01, $n$ = 70). Harvesting at any time later than 50 days after silking did not affect the starch content; the average PM was 70 days after silking. Harvesting prior to 50 days after silking reduced the grain starch content by 0.197% $\mathrm{d}^{-1}$. There were differences in the rate of starch content reduction between the hybrids. For each nutrient quality component, the yield per unit area of grain first increased, then stabilized as the harvest period was delayed. These data could also be fitted using a linear–plateau model. The rate of yield increase varied between the nutritional quality components. Overall, the starch, fiber, and free fatty acids reached maximum yield during the same time period and showed consistent patterns with kernel weight. However, the maximum protein yield was associated with decreased grain MC during harvest. These results indicated that the early-maturing hybrid KWS9384 should be harvested after PM when MC is below 17.5%, whereas the mid–late–maturing hybrid XY335 should be harvested after PM when MC is below 23.6%. When the grain MC of XY335 was lower than 26% (similar to the MC at the PM stage), the contents of starch, oil, and fiber were not affected (Table 4, Figure 5).

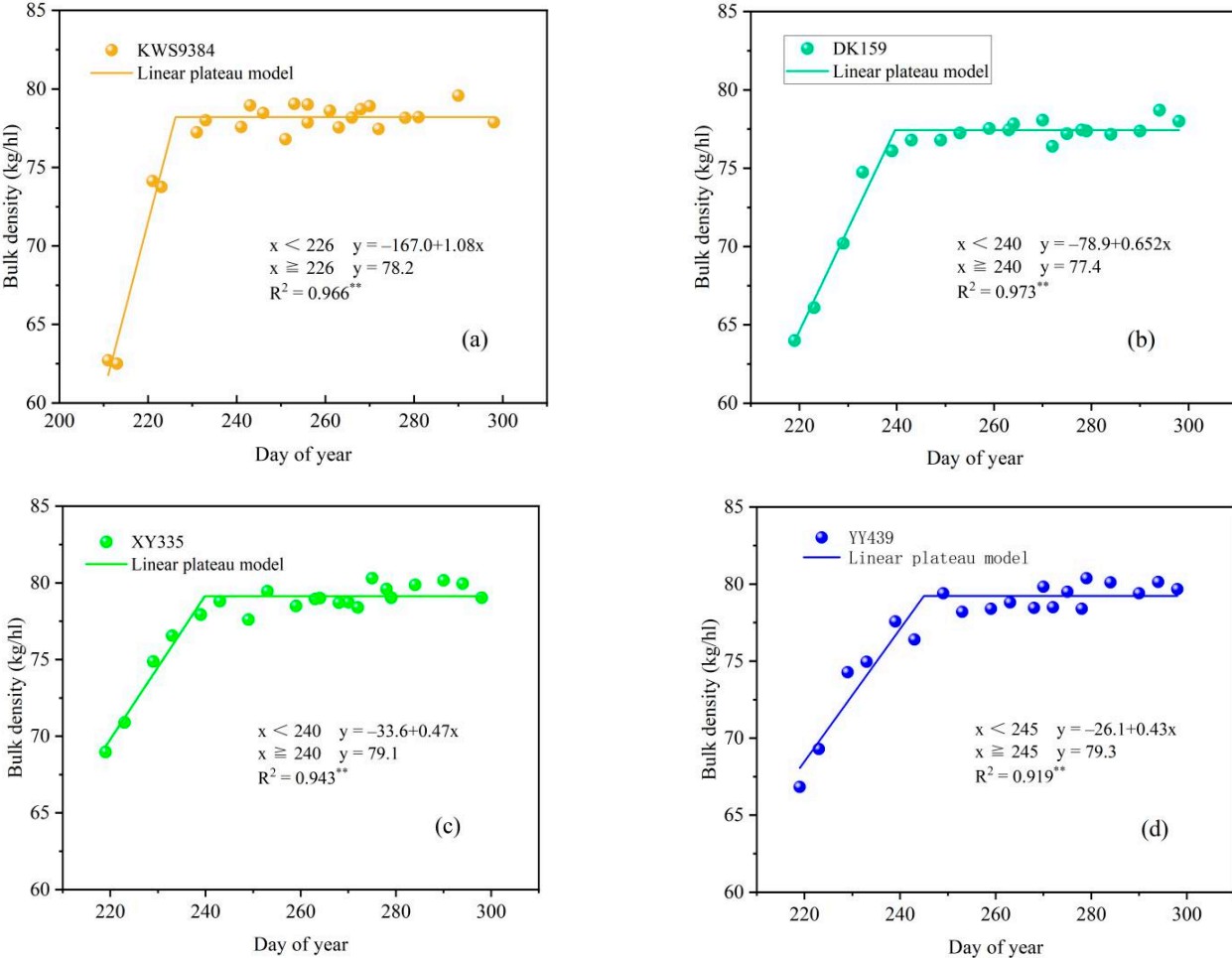

**Figure 4.** Maize bulk density during the harvest period in four hybrids as predicted with linear-plateau models. Values are shown for (**a**) KWS9384, (**b**) DK159, (**c**) XY335, and (**d**) YY439. ** Coefficient of determination (R2) was significant at *p* < 0.01.

**Table 4.** Parameters for linear-plateau models used to analyze the relationships between nutritional component yield and harvest time of hybrids KWS9384 and XY335.

| Nutritional Component Yield | Hybrid | Model Coefficient | | | Join Point | | |
|---|---|---|---|---|---|---|---|
| | | Intercept | Linear Slope | $R^2$ | Days Since Start of Year | Yield (Mg ha$^{-1}$) | MC% |
| Protein | KWS9384 | −1.8 | 0.01 | 0.5885 ** | 263 | 1.3 | 17.5 |
| | XY335 | −2.6 | 0.02 | 0.8322 ** | 267 | 1.5 | 23.6 |
| Oil | KWS9384 | −1.3 | 0.01 | 0.5983 ** | 249 | 0.5 | 23.2 |
| | XY335 | −0.7 | 0.00 | 0.4748 ** | 260 | 0.6 | 26.4 |
| Starch | KWS9384 | −19.2 | 0.11 | 0.7461 ** | 256 | 10.0 | 19.9 |
| | XY335 | −30.6 | 0.16 | 0.9116 ** | 261 | 12.1 | 26.0 |
| Fiber | KWS9384 | −3.3 | 0.02 | 0.5896 ** | 237 | 0.5 | 30.3 |
| | XY335 | −1.4 | 0.01 | 0.7416 ** | 259 | 0.5 | 27.0 |
| Free fatty acids | KWS9384 | −48.9 | 0.23 | 0.5400 ** | 254 | 9.5 | 20.8 |
| | XY335 | −14.8 | 0.09 | 0.2123 | 261 | 9.1 | 26.0 |

** Coefficient of determination ($R^2$) was significant at *p* < 0.01.

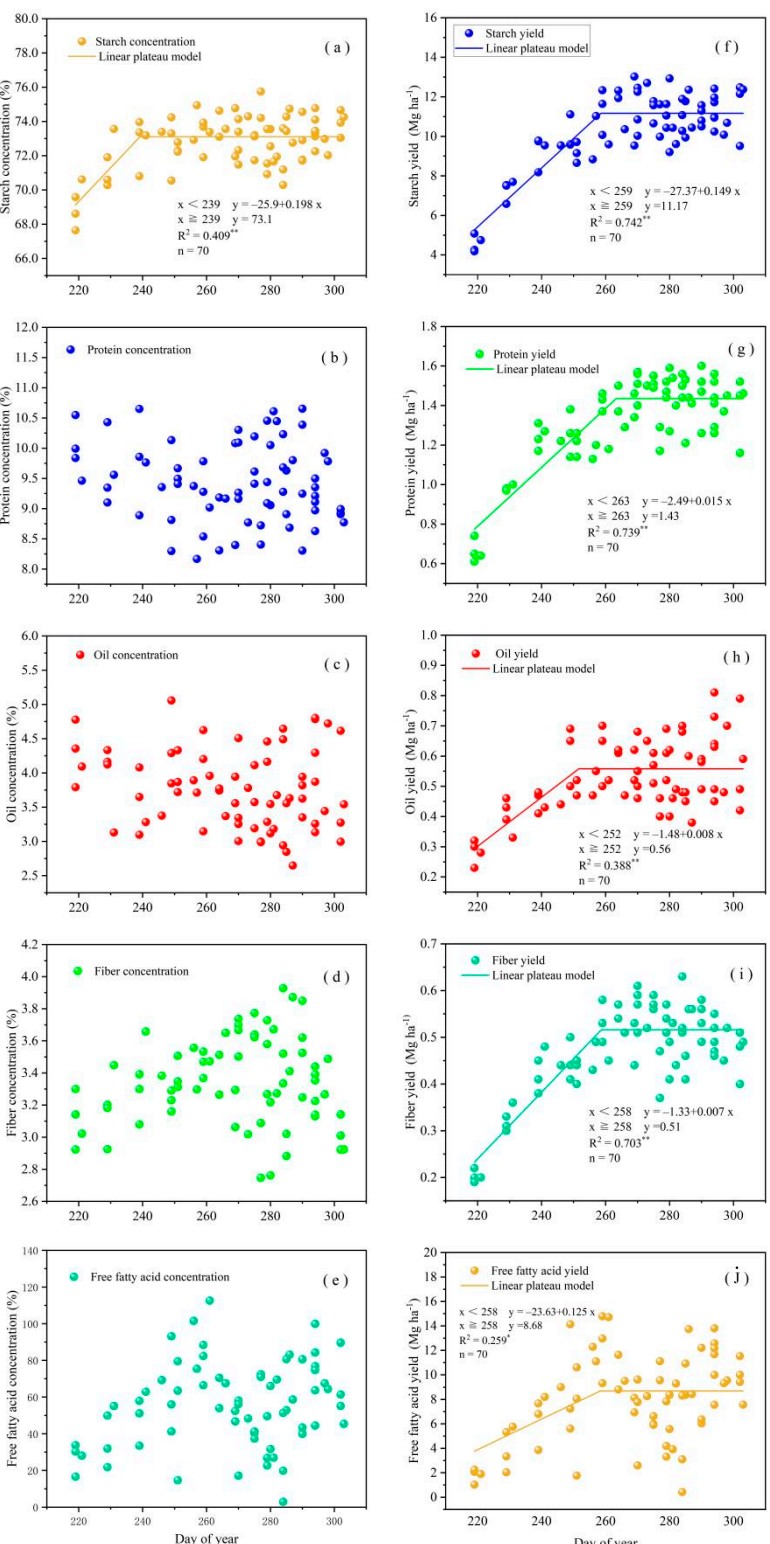

**Figure 5.** Linear–plateau models showing changes in maize kernel protein, starch, oil, fiber, and free amino acid contents of KWS9384, DK159, XY335, and YY439 as the harvest period was delayed. (**a**–**e**), changes in concentrations of each nutritional component; (**f**–**j**), changes in the yield of each nutritional component. ** Coefficient of determination ($R^2$) was significant at $p < 0.01$. * Coefficient of determination ($R^2$) was significant at $p < 0.05$.

## 4. Discussion

It has become common for farmers to reduce maize drying costs by delaying harvest, which reduces grain MC directly in the field [12,14]. Some studies have confirmed that delaying maize harvest for one to two weeks after PM decreases grain MC from 30% to 15%, resulting in yield losses of up to 1255 kg ha$^{-1}$ [20]. Delaying harvest from a high grain MC to an MC of 26% can cause maize yield losses of 10–40 kg ha$^{-1}$ d$^{-1}$ [49]. In the present study, we analyzed dynamic changes in maize grain MC from 30 d after silking through 30 d after PM in the Ningxia Irrigation Area. Rainfall was relatively low during the tested growing seasons (2019–2022), which led to favorable conditions for drying maize stalks and grains directly in the field [11]. From the timepoint with the highest measured grain MC (60.6%) to the timepoint with the lowest MC (11.2%), grain weight first increased, then stabilized. This trend was observed across cultivars with different maturity types. When the grain MC exceeded 31.0% prior to PM, the 1000-kernel weight decreased by 4.44–5.6 g d$^{-1}$; in contrast, when the grain MC was ≤31.0%, the kernel weight and yield remained stable. Research conducted in the main maize production areas in China's Huang-huai-hai Plain has shown that delayed harvesting reduces grain MC without causing yield losses due to decreased grain weight [12], consistent with our results. Hunter et al. [50] found that the same hybrid did not lose any seed dry matter after PM. Paszkiewicz et al. [51] found no changes in kernel dry matter content after PM in the majority of cultivars studied (14 of 18 hybrids in one experiment and 37 of 42 in another). The results of the present study showed that grain weight remained unchanged after PM across four hybrids in four different years. The MC in the harvested grains ranged from 25.9–31.0%, and the impact of MC on grain weight varied between cultivars. The range of grain MC measured here was consistent with previous reports of grain MC in many maize hybrids during the PM period [12,31]. The differences in grain weight between hybrids were not affected by their maturation periods [14,52]. We found that harvesting at any time point after PM did not cause yield loss. However, care must still be taken in mechanical harvesting; yield loss could result from a high machine harvesting fragmentation rate, field loss rate, or lodging rate. Generally, the mechanical harvesting fragmentation rate is lowest when the grain MC is between 18% and 23% [8,33,53,54].

Quality traits are crucial components of maize production. Grain quality is primarily determined during the filling stage [55]. Grain bulk density is a key indicator for evaluating quality and yield [33]. We here found that changes in bulk density caused by differences in harvest time were consistent with changes in grain weight; bulk density first increased, then stabilized. However, grain bulk density stabilized earlier than grain weight did. When maize was harvested at MC < 37%, grain bulk density was not affected. Cloninger et al. [56] found that grain protein concentrations remained unchanged during different harvest periods, but that oil concentrations decreased. Here, we found no significant differences in protein, oil, fiber, or free fatty acid contents in the grains from 30 days after silking to 30 days after PM. Thus, harvesting maize at any time later than 30 days after silking did not affect the protein, oil, fiber, or free fatty acid contents. In contrast, the starch content first increased, then stabilized as the harvest period was delayed. Grain starch content stabilized by 50 days after silking, with the starch contents of grains harvested prior to that timepoint decreasing by 1.97% d$^{-1}$. Previous studies have indicated that delayed harvesting does not affect wheat protein concentrations [57]. Md et al. [14] found that delaying harvest after PM did not change the quality or content of proteins, oils, or starches in the grains; those results were consistent across locations, years, harvest dates, and cultivars. The results of the present study were similar to those of Md et al. [14], with no changes in the starch, protein, oil, fiber, or free fatty acid contents or quality in maize kernels harvested at any time beyond 20 days before PM.

We here found that maize grain bulk density stabilized earlier than grain weight. In the early-maturing hybrid, grain weight (yield) stabilized during PM, whereas in the mid–late-maturing hybrids, grain weight stabilized 5–12 days before PM. Grain bulk density stabilized one to two weeks earlier than grain weight, which was two to three weeks

before PM. However, there were differences between hybrids that matured at different times. The data indicated that harvesting maize two to three weeks before PM would affect yield but not bulk density or quality. Thus, in areas with low AT, or circumstances in which an early frost prevented maize from reaching PM, harvesting mid–late-maturing hybrids at 5–12 days before PM would not affect maize yield or nutritional quality, and harvesting at two to three weeks before PM would not affect bulk density or nutritional quality. However, these plants would have incompletely mature seeds with high MC during harvest, meaning extra care should be taken during mechanical harvesting and grain drying. High maize grain MC is associated with low mechanical harvesting quality, which increases the cost of mechanical operations [8–10,58]. Decreasing grain MC also reduces the risk of mold growth and toxin production in the grains [29,30]. In the literature, differences in yield responses of maize to delayed harvest have been partially attributed to genetic backgrounds and environmental conditions. The selection of maize varieties that are suitable for a specific environment and planting regimen is a critically important management strategy for farmers; the genetic background influences maize maturity time and the susceptibility to stalk lodging before late harvesting [59]. Previous studies have shown that straw lodging caused by delayed harvesting is the main cause of maize yield loss [22,60]. Allen et al. [61] found that when MC was 15% at harvesting, the straw lodging rate increased by 42%, resulting in a yield loss of up to 30%. The irrigation areas of northwest China are rich in light and heat resources, with low air humidity and low rainfall, providing highly favorable conditions for drying maize grains and straw in the field after PM. In recent years, Chinese agricultural researchers have developed a low-moisture, direct-harvest technology based on the delayed and late harvest technologies used for maize in the northwest region. This strategy was identified as the main agricultural technology of China in 2022 (http://www.moa.gov.cn/, accessed on 6 September 2022), and it is therefore of great significance for improving the quality and efficiency of maize production. Combining efficient production technologies with the natural light and heat resources of this region will help to ensure continued high yield and quality in the maize produced in this unique area.

## 5. Conclusions

Maize grain quality is an important economic trait that determines the market price and application value of maize. Field dehydration of maize grains can reduce costs associated with airing and drying after harvest. This study provides the first detailed analysis of changes in grain weight and quality during the transition from high to low grain MC. In the Ningxia Yellow River Irrigation Area, the linear-plateau model results showed that harvesting grains with MC > 31.0% (i.e., before PM) affected maize grain weight, whereas harvesting grains with MC < 25.9% did not affect grain weight or yield. Furthermore, harvesting grains with MC < 37% did not affect grain bulk density, which tended to stabilize one to two weeks earlier than grain weight did (two to three weeks before PM). Early-maturing hybrids showed grain weight stabilization during PM, whereas grain weight stabilized 5–12 days before PM in mid–late-maturing hybrids. Levels of protein, oil, fiber, and free fatty acids in the kernels tended to stabilize around 30 days after silking, but starch formation did not stabilize until 50 days after silking. Delayed harvesting of maize after PM did not affect key qualities such as grain weight, bulk density, or levels of starch, protein, oil, fiber, or free fatty acids. This study serves as an important reference for delaying and improving maize yield and quality during low-moisture harvesting in the heat- and light-rich region of northwest China.

**Author Contributions:** Conceptualization, R.Z., J.G., Y.W. and S.L.; methodology, R.Z., J.G., X.Y., W.L., D.M. and S.L.; software, R.Z.; validation, W.L., X.Y., J.G., B.M. and S.L.; formal analysis, R.Z., J.G., W.L., X.Y., D.M., B.M. and S.L.; investigation, R.Z., Y.W., H.L., W.Z. and Q.C.; resources, Y.W. and S.L.; data curation, R.Z. and H.L.; writing—original draft preparation, R.Z. and Y.W.; writing—review and editing, R.Z., Y.W. and S.L.; visualization, R.Z., J.G. and W.L.; supervision, J.G. and S.L.; project administration, Y.W.; funding acquisition, R.Z. and Y.W. All authors have read and agreed to the published version of the manuscript.

**Funding:** This research was supported by the Natural Science Foundation of Ningxia (2020AAC03306), the High Quality Agricultural Development and Ecological Protection Technology Innovation Demonstration Project (NGSB-2021-3), the Modern Agro-industry Technology Research System in China (CARS-02-80), and the Ningxia Youth Top Talent Training Project (2018).

**Institutional Review Board Statement:** Not applicable.

**Data Availability Statement:** The data are not publicly available due to privacy.

**Conflicts of Interest:** The authors declare no conflict of interest.

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
