# Peer review of "Dynamics of Maize Grain Weight and Quality during Field Dehydration and Delayed Harvesting"

_agriculture, doi:10.3390/agriculture13071357_

Round 1

Reviewer 1 Report

THE REVISED ARTICLE COMPLIES WITH THE SCIENTIFIC RIGOR AND THE APPLIED METHODOLOGY IN ORDER TO PRESENT RELEVANT RESULTS ABOUT THE QUALITY ATTRIBUTES AND THE HARVEST PERIOD EVALUATED

I CONSIDER THAT THE RESULTS PRESENTED AS WELL AS THEIR EXPLANATION ARE INTERESTING FROM THE SCIENTIFIC AND COMMERCIAL POINT OF VIEW

IN THE ATTACHED DOCUMENT YOU WILL FIND OBSERVATIONS THAT SHOULD BE TAKEN INTO ACCOUNT BEFORE BEING ABLE TO PUBLISH YOUR MANUSCRIPT

Reviewer 2 Report

The manuscript completely lack of a statistical description of the analyses carried out by the authors. This paragraph in the Materials and Methods  section must be improved; please clearly explain the models fitted, the factors tested and the tests used. Without this part it is not possible to clearly understand the results and then the discussion. 

Author Response

The responses to reviewer 2 were marked with blue words in the manuscript.

Reviewer 2:

The manuscript completely lack of a statistical description of the analyses carried out by the authors. This paragraph in the Materials and Methods section must be improved; please clearly explain the models fitted, the factors tested and the tests used. Without this part it is not possible to clearly understand the results and then the discussion.

Response: Thanks for your suggestion. We have added the statistical description of the analyses and revised the Materials and Methods section according to your suggestions in the manuscript. (Line 186-192)

Reviewer 3 Report

The topic of the paper is interesting and dealt with important quality attributes of maize grain. In my view, the paper needs revision in the following points:

1. In the abstract part, the author should state a brief about the methodology in one/two sentence/s. 

2. L 32-35: Rewrite this part mentioning the recommendation from your study. Like maize grain should be harvested during ..... days to get better yield with superior quality.

3. In addition to the reference no. 1, 2, and 3, the authors are advised to add the following reference: 1.     https://doi.org/10.3390/agronomy12112766

4. The authors must establish the need of the study in the last paragraph of the introduction. What was the hypothesis? 

5. Experimental site: add some soil data here. 

6. Rewrite the statistical analysis part by mentioning the type of ANOVA, procedures used for mean separation, etc.

7. The results part is unnecessarily lengthy. The authors need to mention the most important findings. The description of the linear plateau graph is not correct. The authors need to recheck this part in the results section. 

8. The discussion part need not be divided into sub-chapters. 

Round 2

Reviewer 2 Report

I have no other comments or suggestion to make, the manuscript now is well written and also the statistical part is clear.

Reviewer 3 Report

The authors have improved the article as per the recommendation given. Reference 4 is not properly written in a bibliographic manner. It should be written as Surname, Name.; Surname, Name.; and so on. Make the minor correction accordingly.

In my view, the article may be accepted now.